# CM-Dil Staining and SEC of Plasma as an Approach to Increase Sensitivity of Extracellular Nanovesicles Quantification by Bead-Assisted Flow Cytometry

**DOI:** 10.3390/membranes11070526

**Published:** 2021-07-13

**Authors:** Nadezhda Nikiforova, Maria Chumachenko, Inga Nazarova, Lidia Zabegina, Maria Slyusarenko, Elena Sidina, Anastasia Malek

**Affiliations:** 1Subcellular Technology Laboratory, N.N. Petrov National Medical Center of Oncology, 197758 St. Petersburg, Russia; niki2naden_ka@mail.ru (N.N.); oblaka12@mail.ru (I.N.); lidusikza@yandex.ru (L.Z.); slusarenko_masha@mail.ru (M.S.); sidina@mail.ru (E.S.); 2Department of Biochemistry, Belarusian State University, 220030 Minsk, Belarus; chumachenkomaria19@gmail.com; 3The Faculty of Physics, Saint-Petersburg State University, 199034 St. Petersburg, Russia; 4Oncosystem Ltd., 121205 Moscow, Russia

**Keywords:** extracellular vesicles, plasma, CM-Dil, labeling, size exclusion chromatography, SEC, flow cytometry, FCM

## Abstract

The quantification of the specific disease-associated populations of circulating extracellular membrane nanovesicles (ENVs) has opened up new opportunities for liquid biopsy in cancer and other chronic diseases. However, the sensitivity of such methods is mediated by an optimal combination of the isolation and labeling approaches, and is not yet sufficient for routine clinical application. The presented study aimed to develop, characterize, and explore a new approach to non-specific ENV staining, followed by size-exclusive chromatography (SEC), which allows us to increase the sensitivity of bead-assisted flow cytometry. Plasma from healthy donors was purified from large components, stained with lipophilic CM-Dil dye, and fractionated by means of SEC. The obtained fractions were analyzed in terms of particle size and concentration using NTA, as well as vesicular markers and plasma protein content via dot-blotting. We characterized the process of CM-Dil-stained plasma fractionation in detail and indicated the fractions with optimal characteristics. Finally, we explored the sensitivity of on-bead flow cytometry for the analysis of specific populations of plasma ENVs and demonstrated the advantages and limitations of the proposed technique.

## 1. Introduction

Plasma-circulating membrane extracellular nanovesicles (ENVs) have great diagnostic potency, especially in oncology [1]. The development of new ENV-based diagnostic approaches is challenging due to their small size and heterogeneity. Since the diagnostic purpose assumes the quantification of the particular disease-indicative ENV sub-populations, these vesicles need to be isolated, labeled, and then calculated. The current approaches to ENV isolation [2] and further analysis [3] are quite diverse and comprehensive. Moreover, the method of ENV isolation strongly affects the results of further analyses [4], which leads to discrepancies. To solve this issue, the International Society for Extracellular Vesicles (ISEV) provides detailed guidelines [5] and endorses the use of web platforms (EV-TRACK, EV-METRIC) [6]. Both measures are intended to standardize protocols and enable data comparisons across different laboratories. However, the translation of ENV research “from bench to bedside” and the development of clinically usable ENV-based technologies brings to the fore issues related to the simplification and cheapening of analytic protocols. These aspects were addressed during a workshop on EV-based Clinical Theranostics initiated by ISEV in 2018 [7]. Our basic goal was to create a robust and clinically usable method to quantify plasma ENVs bearing specific surface markers by means of flow cytometry. Keeping this purpose in mind and analyzing the relevant publications, we made several considerations regarding the optimal workflow of the analytic procedure.

The first issue is the selection of an optimal approach to flow cytometry (FCM). Currently, two methods are available: conventional FCM, wherein an instrument detects artificial beads with ENVs attached to the surface, and so-called high-resolution FCM (hrFCM), wherein an instrument is used for the detection of individual ENVs. The opportunity for the multi-parametric analysis of individual particles is an obvious advantage of hrFCM. However, this approach involves high requirements with regard to the sensitivity of the device and the skill of the performer. For instance, recently published guidelines (MIFlowCyt-EV, Minimum information about FCM experiments with ENVs) suggest the reporting of 25 experimental components for the confirmation that the correct experimental conditions have been reached [8]. The alternative approach—bead-assisted FCM—is much more reliable in terms of its technical performance; however, it usually requires the immune-capturing of ENVs on the surface of micro-sized beads. Since this technique was established [9,10], various modifications of the procedure have been successfully explored [11,12,13] to make it clinically viable.

Another concern is the method of fluorescent labeling of on-bead ENVs. In the above-cited reports, ENVs were labeled with antibodies specific to markers of interest. However, this approach assumes double-selection of the vesicles to be quantified: first by immune-capturing, and then by immune-labeling. This technique can be useful for experimental research, but can produce a biased clinical interpretation. Moreover, antibodies might have a relatively high cost, low stability, and batch-to-batch variability. Therefore, the use of only one type of marker-specific antibody (either for capturing or for detection) would decrease the cost and increase confidence in the results. According to this consideration, combining immune-capturing and the non-specific labeling of vesicular membranes might present an optimal strategy. Non-specific labeling of ENVs with lipophilic dyes provides many advantages, including a relatively low cost and high efficacy. However, the common disadvantages, such as micelle formation [14,15] and the labeling of non-vesicular plasma components [16], require attention. An elegant strategy to overcome the inherent limitations of lipophilic compounds was demonstrated recently by J. Maia et al. [17]. The authors of that study stained plasma with CFSE, fractionated it via size exclusion chromatography (SEC) to separate stained vesicles from other plasma components and micelles, and analyzed the stained vesicles using hrFCM.

Considering the general goals of clinical applicability and argumentation, as described above, we utilized a pre-analytic workflow, including the labeling of plasma with a lipophilic dye, followed by SEC (Figure 1). Next, the specific populations of plasma ENVs were quantified in selected SEC fractions by means of bead-assisted FCM. To investigate the process of plasma component separation in detail, the SEC-separated fractions were assayed in terms of their particle size, quantity, and vesicular marker content. The results obtained via FCM demonstrated the good performance of the proposed pre-analytic approach and the exceptional sensitivity of the subsequent bead-assisted quantification of specific ENV populations.

## 2. Materials and Methods

### 2.1. Plasma Sampling and Processing

Plasma was obtained from healthy donors after they signed a voluntary and informed study participation form. Blood was collected in BD Vacutainer spray-coated EDTA tubes; plasma was immediately separated from the blood, aliquoted, and stored at −80 °C. Before use, plasma was slowly thawed at 4 °C. In order to remove cells and cellular detritus, plasma was centrifuged at 400× *g*—10 min, 800× *g*—10 min, 1500× *g*—10 min, and 17,000× *g*—30 min. Each time, the supernatant was carefully replaced in a new tube. After the last centrifugation, the supernatant was filtered through a 0.2 mkm PES syringe filter to obtain pellet pure plasma (PPP). Plasma treated in this way was used in all experiments described in this study.

### 2.2. Isolation of ENVs via Differential Ultracentrifugation

PPP (2 mL) was mixed with PBS in a 1:1 ratio. ENVs were sedimented by means of ultracentrifugation (UC) according to the classical procedure, with minor modifications, using an Optima XPN 80 ultracentrifuge (rotor 70.1 Ti/k-factor 36). Briefly, the 50% plasma solution was centrifuged at 110,000× *g* for 2 h, the supernatant was removed, and the precipitate was dissolved in PBS. The solution was then centrifuged again at 110,000× *g* for 2 h. The supernatant was then removed, and the pellet was resuspended in 500 μL of PBS. To isolate the pure cancer cell-derived ENV population, a colon cancer cell culture (Colo320) was cultured under standard conditions in RPMI medium with 10% FCS and penicillin–streptomycin until confluence was reached at 50%–60%. Then, the medium was replaced by fresh serum-free medium and cells were cultured for 2–3 days until they reached 90%–100% of the confluence layer. The medium was collected; centrifuged at 800× *g*—15 min, 1500× *g*—15 min, and 17,000× *g*—1.5 h; and then filtered through a 0.22 mkm PES filtration system. ENVs from the 180 mL pre-treated medium were sedimented by means of ultracentrifugation at 110,000g for 4 h, the supernatant was removed, and the precipitate was dissolved in PBS. The solution was centrifuged again at 110,000× *g* for 2 h. The supernatant was removed, and the pellet was dissolved in 500 μL of PBS.

### 2.3. Transmission Cryo-Electron Microscopy (Cryo-TEM)

The morphology analysis of ENVs isolated from plasma via UC was performed with a Jeol JEM-2100 microscope at the Research Resource Center for Molecular and Cell Technologies of Saint-Petersburg State University. The samples of the ENVs at a concentration of 10^12^ particles/mL were deposited on a carbon-coated copper mesh/Lacey Carbon Supported Copper Grids, 50 nm in size (Sigma-Aldrich, St. Louis, MO, USA). The excess sample was removed with filter paper. After that, the sample was immersed in liquid ethane for rapid freezing and transferred to a cryostat for subsequent analysis using a cryo-microscope.

### 2.4. CM-Dil Plasma Labeling and SEC Purification

An amount of 1 mg/mL (1 µM) of a stock solution of CellTracker CM-Dil reagent (C7001, Thermo Fischer Scientific, Walthman, MA, USA) in 100% DMSO was prepared via ultrasonification (35 kHz, 10 min) and stirring until the dye was completely dissolved. The stock solution was diluted 1:20 and 1:400 to obtain 50 nM and 2.5 nM solutions in DMSO. Then, 2 mkl of the prepared CM-Dil solutions were added to 2 mL PPP, and incubated for 20 min at 37 °C with moderate stirring. Thereafter, the samples were immediately uploaded onto prepared (washed with PBS) SEC columns (HBM-PEV-10, HansaBioMed Life Sciences Ltd., Tallin, Estonia). Twenty-three fractions of 500 μL were collected and analyzed.

### 2.5. Nanoparticle Tracking Analysis

The measurements were performed using the Nanosight NS300 analyzer (Malvern Panalytical, Malvern, UK). In order to analyze ENVs isolated via ultracentrifugation, samples were diluted in ratios of 1:100 and 1:1000, and each dilution was studied in 4–5 different micro volumes by pumping the sample through a chamber. The duration of each measurement was 30 s. Camera level: 14. Shutter slider: 1259. Slider gain: 366. Repeated results, obtained with the optimal dilution, were used after averaging.

In order to measure the size and the concentration of particles distributed by SEC, the following fractions of plasma (PL) and CM Dil stained plasma (PL-Dil) were assayed: combined 5 and 6 fractions, 7, 8, 9, 10, 11, 12, 13, 14, 15, and combined 16–21. The SEC fractions of CM Dil solution were combined in five pools: 5–6, 7–8, 9–11, 12–15, and 16–21. Fractions 1–4 and 22–23 were excluded from the analysis as they were non-informative. All samples were diluted in the ratio of 1:20 by means of PBS before measurement. Camera level: 12–15 for pellet pure plasma and CM Dil-stained plasma, 15–16 for CM Dil solution. Slider shutter: 1200–1300. Slider gain: 175–295. Number of frames: 749. Detection threshold: 17 for plasma and CM Dil-stained plasma; 7 for CM Dil solution. Each measurement was carried out 6 times for 30 s and the results were averaged. The spectra were processed using Nanosight NTA 3.2 software. The results were transferred in an Excel table. Similar results obtained using NTA of plasma (PL) and CM Dil-stained plasma (PL-Dil) were combined, averaging them into pools of 7–8, 9–11, and 12–15 to be presented graphically. The raw data are available in the Appendix A.

### 2.6. Fluorescent Intensity Assessment and Dot-Blotting

The 0.3-µL samples obtained via SEC ware spotted onto the nitrocellulose membrane. The fluorescence intensity of the CM-Dil dye was measured on an Invitrogen iBright FL1500 Imaging System with an exposure time of 0.016 s. The values were digitized using the ImageJ program. To assay the protein content of SEC-separated plasma fractions, the membrane with spotted samples was dried, blocked by soaking in 5% BSA in TBS-T in RT for 30 min, and incubated at 4 °C overnight with primary antibodies against CD9 (HBM-CD9-100, HansaBioMed, Tallin, Estonia), CD63 (ab68418, Abcam, Cambridge, UK), HSP70 (kindly provided by Dr. A. Zhahov, RU patent 2 722 398), HSA (4T24, HyTest, Turku, Finland), Reg IV (ABP56724, Abbkine, Wuhan, China), and DHRS11 (ABP56569, Abbkine, Wuhan, China). Primary antibodies were dissolved to 1 mkg/mL in 0.1% BSA in TBS-T. Then, the membrane was washed three times with TBS-T (3 *×* 5 min), incubated at 4 °C for 30 min with secondary HRP-conjugated Fc fragment-free antibodies (ab6823, ab7171, AbCam, Cambridge, UK), diluted at a ratio of 1:25,000 in 0.1% BSA in TBS-T, and washed three more times with TBS-T (15 min × 2 times), then once with TBS (5 min). The membrane was soaked in Immobilon Classico Western HRP substrate (WBLUC0020, Millipore, Burlington, MA, USA) in the dark for 5 min. The images were obtained using the Invitrogen iBright FL1500 Imaging System.

### 2.7. Immunobeads Fabrication and On-Beads FCM

The pellet pure plasma (2 mL) was stained with CM-Dil (2.5 pM) and SEC fractionated. For the quantification of colon cancer-derived ENVs, plasma was mixed with a corresponding amount of COLO320-derived vesicles before being stained with dye. SEC fractions 9, 10, and 11 were pooled and 100 µL of each pooled sample was incubated with 0.07 µg of the biotinylated antibodies—HSP70 (kindly provided by Dr. A. Zhahov, RU patent 2 722 3980), DHRS11 (ABP56569-biotin, Abbkine, Wuhan, China), and Reg IV (ABP56724-biotin, Abbkine, Wuhan, China)—overnight at 4 °C with gentle stirring. Superparamagnetic beads (SPMB) coated with streptavidin (K0180, Sileks Ltd., Moscow, Russia) were washed three times with PBS and blocked with 0.2% Protein-Based Blocking Reagent (T2015, Invitrogen, Walthman, MA, USA) for one hour at 4 °C with gentle stirring. Pre-treated SPMB (1 µg) were gently mixed with ENVs coupled with biotinylated antibodies for one hour at 4 °C. The obtained complexes were washed three times to rid them of excessive ENVs. In order to label CD63 markers on the surface of on-bead-attached ENVs, complexes were additionally incubated with FITC-conjugated, mouse monoclonal MEM-259 against CD63 (AbCam, Cambridge, UK) and washed again. Finally, the SPMB-ENVs complexes were resuspended in 200 µL of PBS and analyzed using CytoFLEX. Data processing was performed with the FlowJo™ v10.7 software.

## 3. Results

### 3.1. Design of Study

A goal of the study was to estimate whether CM-Dil plasma staining followed by SEC presents an efficient approach for on-bead FCM of plasma ENVs. With this purpose, three objectives were addressed (Table 1).

First, we aimed to evaluate the efficacy of plasma component labeling with CM-Dil. Two samples—pellet pure plasma (PPP) and plasma ENVs isolated by ultracentrifugation (ENVs)—were included in this experiment. SEC fractions from 5 to 23 were analyzed. Next, we planned to characterize particles that may be presented in CM-Dil-labeled plasma; CM-Dil stained PPP was compared with PPP and CM-Dil solution in PBS. SEC fractions from 5 to 21 were analyzed. Finally, we planned to evaluate the efficacy and sensitivity of on-bead FCM of CM-Dil-stained and SEC-purified plasma ENVs in different experimental settings. Fractions 9, 10, and 11 enriched by ENVs were included in analysis. The corresponding methods and samples used are indicated in Table 1.

### 3.2. Isolation and Characterization of Plasma ENVs

In order to evaluate the efficacy of CM-Dil labeling of different plasma components, ENVs were isolated from pellet pure plasma (PPP) via two rounds of ultracentrifugation, as described. The concentration of vesicles was measured via NTA and adjusted to 10^12^ particles/mL. Vesicles were visualized using Cryo-TEM (Figure 2). The cluster of vesicles is visible in the left panel of Figure 2, and an enlarged image of a single vesicle is presented in the right panel. The results of Cryo-TEM confirmed the vesicular structure of the isolated particles. Thus, the vesicles isolated by UC were used in next experiments as a control.

### 3.3. Efficacy of Plasma Component Labeling with CM-DIl

The protocol we used as a prototype [17] included the staining of plasma with CFSE membrane-permeable dye. In our study, we used the lipophilic compound CM-Dil, which is expected to interact with various lipid-containing plasma components, as well as vesicles. To evaluate the presence and amount of non-vesicular dye-binding substances, pellet pure plasma (2 mL) and ENVs (isolated via ultra-centrifugation from 2 mL of pellet pure plasma) were stained with CM-Dil (1000 pM), fractionated by SEC, spotted onto a nitrocellulose membrane, and evaluated for fluorescent signal intensity. Figure 3A presents the results of the fluo-dot-blot analysis of SEC fractions 7 to 23, and the data quantification. As expected, UC-isolated ENVs were mostly detected in fractions 8, 9, 10, and 11; these vesicles were intensively stained with fluorescent dye. Other SEC fractions were free of dye-stained components. In contrast, plasma contained a broad spectrum of CM-Dil-stained components that were distributed between fractions 8 and 23. Here, we detected the smallest membranous structures and lipoproteins presented in the plasma. To compare the affinity of CM-Dil binding exhibited by different plasma components, we stained the plasma with different amounts of dye (1000 pM, 50pM, 2.5 pM) and assayed the intensity of the fluorescent signals. Small lipophilic plasma components turned out to be less attractive to the dye; the profile of fluorescently labeled plasma components became similar to that of UC-isolated ENVs at a CM-Dil concentration of 2.5 pM (Figure 3A). This result allowed us to establish an optimal CM-Dil concentration (2.5 pM) that resulted in the predominant staining of plasma ENVs.

### 3.4. Size and Concentration of SEC-Fractionated Plasma Particles

As indicated in a number of reports, lipophilic dyes such as CM-Dil may form micelles that are similar in size and structure to ENVs. As these micelles may interfere with the further analysis of CM-Dil-stained plasma ENVs, we intended to track their formation. The aim of our next experiment was to evaluate the size and concentration of components of SEC fractions obtained using separation plasma (PL), CM-Dil-stained plasma (PL-Dil), and CM-Dil alone. The latter two samples contained the same amount of dye (2.5 pM). After performing SEC, fractions 5 and 6, 7 and 8, 9–11, 12–15, and 16–21 were combined and measured by means of NTA. The results are presented in Figure 4. Thus, SEC fractions 5-6 of plasma did not contain any measurable particles (Figure 4A, yellow color). As we used pellet pure plasma depleted by large components, this result was predictable. The next fractions, including 7 and 8 and 9–11, contained a similar amount (~1 × 10^9^/mL) of particles, at around 100 nm in size. These results reflect the predictable presence of ENVs and are consistent with the fluo-dot-blot results (Figure 3A). The next fractions (12–15) contained fewer smaller particles. However, the last measurement, combining SEC fractions 16–21, revealed the presence of small particles in an amount two orders higher (2.5 × 10^10^/mL) than the amount of ENVs (see the graph for details). We can suppose that these fractions included plasma lipoproteins and protein complexes big enough to be detected by NTA. The CM-Dil-stained plasma appeared to be quite similar to plasma (Figure 4B, red color), fractions 5 and 6 did not contain any detectable particles, and the next two measurements revealed particles of ENV size (100 nm) in concentrations from 0.5 × 10^9^/mL to 1.2 × 10^9^/mL. It can be noted, via a comparison of the results of the plasma and CM-Dil-stained plasma analyses, that the number of relatively heavy particles (fractions 7 and 8) decreased, whereas the number of lighter particles (fractions 12–15) increased due to CM-Dil staining. This shift might reflect the change in particle density due to the incorporation of CM-Dil. Moreover, in both samples (plasma and CM-Dil-stained plasma), fractions 9–11 included a similar number of 100-nm particles, which were mostly represented by plasma ENVs (as shown in Figure 3A). The last sample, combining SEC fractions 16–21, again contained particles that were much smaller than 100 nm in a concentration of 5∗10^10^ (as shown in the graph). As the difference in these fractions between the analyzed samples (plasma VS. CM-Dil-stained plasma) was mediated by the presence of the dye only, a twofold increase in the light particle number in SEC fractions 16-21 might be the result of the formation of new complexes by CM-Dil and small plasma components. The results of the pure CM-Dil fractionation are presented in Figure 4C (violet color). The profile of the particle size distribution of the dye suspension differed considerably from the plasma; fractions from 5 to 15 contained a smaller number (6 × 10^8^–8 × 10^8^/mL) of particles, with a size of about 60–70 nm. These particles may simply have been micelles formed by CM-Dil, as they had a rather different density. The presence of heavy particles in SEC fractions 5 and 6, as well as the absence of light particles in SEC fractions 16–21, were the main differences between the suspension of CM-Dil-formed micelles and the plasma samples. It can be noted that the heavy particles fractioned in fractions 5 and 6 can be formed by CM-Dil in the absence of plasma only; the presence of plasma somehow prevents the formation of such particles. Moreover, an analysis of the CM-Dil suspension confirmed our previous conclusion regarding the relatively light and small components of plasma observed in fractions 16–21; these particles existed in plasma samples only. This experiment allowed us to compare the particle size/density distribution in plasma, CM-Dil-stained plasma, and the CM-Dil suspension and to identify the SEC fractions enriched by ENV-sized particles and minimally contaminated by CM-Dil-formed micelles.

Figure 5 presents the same data set, but allows us to easily compare the amount of particles of similar densities in certain SEC fractions of the studied solutions. It can be clearly seen that fractions 5 and 6 contained particles in the CM-Dil suspension only (Figure 5A). In the next fractions (fractions 7 and 8; Figure 5B), plasma and CM-Dil-stained plasma contained particles of the same vesicular size (100 nm); however, the presence of the dye reduced their number. In fractions 9–11, the number of particles of vesicular size (100 nm) in samples of plasma and CM-Dil-stained plasma was almost the same (Figure 5C). In the next fractions, fractions 12–15 and 16–21, both plasma and CM-Dil-stained plasma contained smaller particles (70–80 nm), and the number of these particles was about twofold higher in CM-Dil-stained plasma samples (Figure 5D,E), which may indicate the causal role of dye in their formation.

Taken together, the results of the NTA reveal that the number/density of plasma particles of vesicular size (100 nm) was less influenced by staining with CM-Dil in fractions 9–11: these fractions, in CM-Dil-stained plasma, most likely include ENVs that are suitable for further analysis. In the next fractions, the size of particles became smaller in both plasma and CM-Dil-stained plasma samples, whereas staining with CM-Dil increased the particle number. These fractions in the CM-Dil-stained plasma most likely included particles of a non-vesicular nature and their analysis requires further attention. Fractions 5 and 6 and 16–21 should be excluded from further analysis because they may contain CM-Dil-formed micelles (in 5 and 6) and small plasma components (in 16–21).

### 3.5. Protein Content of SEC Fractions of Plasma and CM-Dil-Stained Plasma

As is well known, different vesicular markers may be expressed in different vesicular populations. Apparently, these populations may be separated by means of SEC. We used dot-blotting for the relative quantification of vesicular markers CD63 and CD9, as well as abundant plasma proteins HSP70 and albumin in different SEC fractions of plasma. To explore the possible influence of CM-Dil staining, plasma and CM-Dil-stained plasma were assayed in parallel. The results of the CD63 analysis are presented in Figure 6A, and the results of the complete data quantification are presented in Figure 6B. Thus, vesicular marker tetrasponin CD63 in SEC-separated plasma was detected in fractions 10 to 20, with maximum intensity in fractions 14–16. Tetrasponin CD9 was less abundant, but had a similar profile of distribution throughout the SEC fractions. These data indicate that the maximal concentration of CD63/CD9-positive vesicles (so-called exosomes) can be found in fractions 14–16. HSP70 was detected mostly in fractions 12 to 16, thereby potentially confirming the presence of this protein in vesicular form rather than in soluble form. A low concentration of albumin was already detected in fractions 12 and 13. The quantity of albumin increased in the subsequent fractions; however, the method of detection reached its saturation point at fraction 15, and further increases in albumin concentration cannot be estimated. Thus, ENV samples are contaminated with albumin and apparently with other small plasma proteins in fractions 13 and 14 and subsequent SEC fractions; the possible impact of abundant plasma proteins on the efficacy of further analysis of ENVs from these fractions must be considered.

The results of dot-blotting also revealed an interesting effect of CM-Dil staining. Certain shifts in the blot signal intensity of later fractions were observed through a comparison of the plasma and CM-Dil-stained plasma samples. This may happen due to the incorporation of lipophilic dye into lipid-containing plasma components, resulting in a change in their density.

By combining the results of the NTA and the dot-blotting, we can conclude that fractions 9–11, obtained by means of SEC of the CM-Dil-stained plasma, contained vesicles of 100 nm, with low CD63/CD9 representation, whereas fractions 12–16 contained vesicles of a smaller size; however, they were also CD63/CD9-positive.

### 3.6. Immunocapturing and Bead-Assisted FCM

Next, we intended to evaluate the efficacy of the developed method of ENV labeling as a pre-analytic step in bead-assisted FCM. Super-paramagnetic beads were decorated with antibodies against HSP70 via a biotin–streptavidin link. Pellet pure plasma was stained with CM-Dil and SEC fractionated. Fractions 9–11 were combined and incubated with SPMB. Afterwards, SPMB with coupled HSP(+) ENVs were stained with FITC-labeled antibodies against CD63. Flow cytometry was performed to assay the signal intensity from both the antibodies (FITC channel, 525 nm) and CM-Dil (PE channel, 575 nm). The scheme of the experiment and the results are shown in Figure 7. Thus, the proposed approach to ENV labeling allowed us to detect 9.46% of counted beads as PE positive, whereas FITC-positive beads only accounted for 4.52%.

As fluorescent signals in two different channels were obtained from the same sample, the observed difference in the fluorescence intensity could only be caused by the different efficacies of ENV labeling. Thus, the non-specific labeling of the vesicular membrane with lipophilic CM-Dil dye turned out to be two times more efficient compared to the conventional method of common vesicular marker labeling with FITC-coupled antibodies.

Considering potential clinical uses, the quantification of low-representative disease-associated ENV populations would be the most important application of the developed method. As the efficacy of CM-Dil labeling is relatively high, it is expected to mediate the high sensitivity of bead-assisted FCM. The final experiment aimed to prove this assumption using model mixtures of ENVs.

As shown previously, the development of colon cancer is associated with an increase in the concentration of circulating ENVs with intestinal-specific surface markers [18]. Two proteins specifically expressed in the intestinal epithelium were selected for the next experiment: REG-IV and DRHS11. ENVs isolated from the conditional media of colon cancer cells (COLO320) were tested for the presence of these proteins as a suspension of intact vesicles and as a lysate using the dot-blot method (Figure 8A). The results revealed that these proteins were expressed predominantly in the vesicular membrane, alongside CD63, which was tested as the control. Therefore, SPMB decorated with antibodies against these proteins can be used for the isolation of specific REG-IV(+) ENVs or DRHS11(+) ENV populations. COLO320-derived vesicles were isolated by means of ultra-centrifugation, quantified by means of NTA, and an increasing quantity of these vesicles (10^9^, 10^10^, 10^11^, and 2 × 10^11^) was mixed with pellet pure plasma (2 mL). The mixtures were CM-Dil-stained and SEC fractionated. Fractions 9–11 were incubated with SPMB decorated with antibodies against proteins of interest. As in previous experiments, SPMB with coupled REG-IV(+) ENVs or DRHS11(+) ENVs were stained with FITC-labeled antibodies against CD63 and the intensity values of the FITC and PE channels were measured by means of FCM. No CD63(+) events were detected in the FITC channel, whereas the increased signal intensity was evaluated through an analysis of the CM-Dil-stained ENVs in the PE channel (Figure 8B). As the concentration of ENVs in plasma was estimated to be in the range of 10^12^–10^13^/mL, the quantity of COLO320-derived vesicles could be approximated as 1%, 0.5%, 0.05%, and 0.005% of the total number of vesicles in the mixture. These results demonstrated the extremely high sensitivity of the proposed analytic approach.

## 4. Discussion

In the present study, we developed a new approach to plasma ENV labeling and isolation that can be used to prepare vesicles for subsequent quantification by means of bead-assisted FCM. The proposed technique mediates effective labeling because of the non-specific staining, with lipophilic dye, of the vesicular membrane, whereas the subsequent use of size-exclusive chromatography allows us to separate stained ENVs from the non-informative plasma components and micelles occasionally formed by dye. We have demonstrated that excess dye can be bound by various other plasma components; however, the staining of vesicles is more effective and requires a lower concentration of dye. We investigated the SEC-mediated separation of plasma components in terms of particle size and concentration in detail, as well as their contents of vesicular markers and major plasma proteins. The obtained results revealed that SEC fractions 9–11 contained a maximal number of 100-nm particles, which most likely represented exosomes. The next fractions (12–16) contained particles of a smaller size; however, these were enriched with exosomal markers CD63, CD9, and HSP70. An increasing amount of albumin was detected, starting from fractions 11 and 12. Interestingly, CM-Dil staining resulted in a slight shift in the size and density of plasma particles detected by both NTA and dot-blotting. Thus, our results provided a set of characteristics to assist with the process of size exclusion chromatography in plasma and CM-Dil-stained plasma.

By maintaining the general workflow of the procedure, the proposed technique could be easily modified by using different lipophilic or membrane-incorporating dyes and different methods of size-exclusive chromatography. The list of membrane-labeling compounds is constantly growing [19,20] and includes PKH26, PKH67 [14], the water-soluble dyes DiD, Dio, DiR, and CM-Dil [21], the membrane-permeant molecule calcein AM [22], the amine-reactive dye CFSE [23], and two-component molecules with fluorescence polarization properties such as C12-FAM [24]. Thus, all of these dyes interact non-specifically with the vesicular membrane and can mediate ENV labeling, which is effective enough to enable the sensitive quantification of small disease-associated vesicular populations. The proposed technology can be scaled up and applied to on-bead FCM for scientific and clinical diagnostic purposes.

Due to its exceptional sensitivity, the proposed approach can be used to analyze a minor fraction of circulating plasma ENVs. It is now assumed that the population of plasma ENVs is extremely heterogenous. It appears to contain vesicles derived from blood cells, the endothelium, and other tissue. However, the exact tissue origin of circulating ENVs has not yet been investigated, nor has the physiological composition of the plasma vesicular population. It may be supposed that changes in the vesicular composition of plasma may reflect various diseases and may serve as important diagnostic criteria. Moreover, the development of a well-differentiated tumor may be associated with an increased amount of specific tissue-derived ENVs in plasma. The sensitive detection of subtle changes in plasma vesicular composition could provide a new approach to the early diagnosis of cancer. The sensitive detection of tumor-derived ENVs with a specific cancer-associated pattern of surface markers can be studied as a method of disease monitoring during the course of therapy. Thus, the proposed technology is sensitive enough to serve as a new tool for accurate plasma ENV analysis and is fairly easy to use in clinical practice.

## Figures and Tables

**Figure 1 membranes-11-00526-f001:**
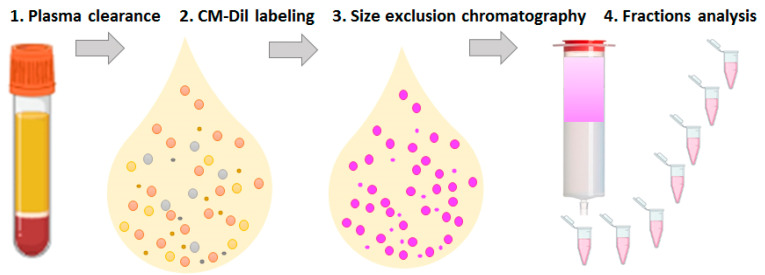
Schematic workflow of the pre-analytic procedure, including the following steps: 1. Clearance of plasma from cells and cellular debris via consequent centrifugation; 2. Labeling of the lipid-containing plasma component by staining plasma with CM-Dil; 3. Fractionation of CM-Dil-stained plasma via size-exclusion chromatography; 4. Analysis of separated fractions.

**Figure 2 membranes-11-00526-f002:**
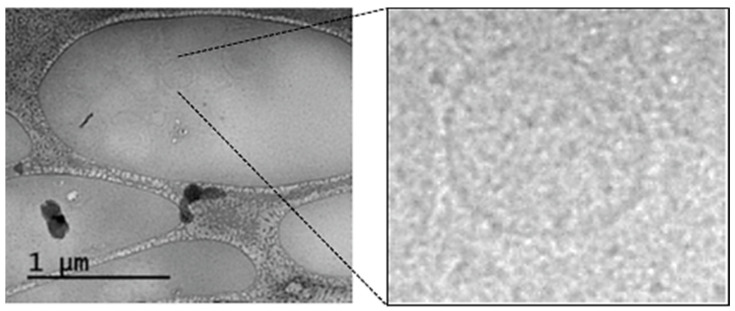
Cryo-transmission electron microscopy of ENVs isolated from plasma by means of UC.

**Figure 3 membranes-11-00526-f003:**
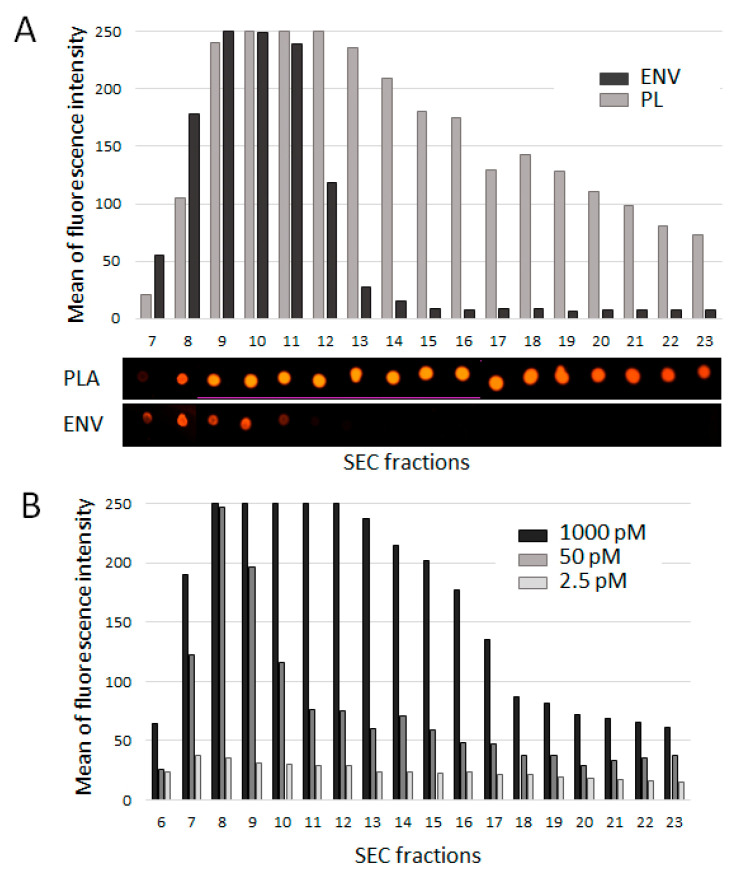
Efficacy of plasma component labeling with CM-Dil. (**A**) Plasma (2 mL) and ENVs isolated from 2 mL plasma were stained with CM-Dil (1000 pM), SEC fractioned, and assayed be means of flu-dot-blot analysis. The results were quantified using an Invitrogen iBright FL1500 Imaging System. (**B**) Plasma samples (2 mL) were stained with CM-Dil at concentrations of 1000 pM, 50 pM, and 2.5 pM; SEC fractionated; and assayed by means of fluo-dot-blot analysis. The results were quantified using an Invitrogen iBright FL1500 Imaging System.

**Figure 4 membranes-11-00526-f004:**
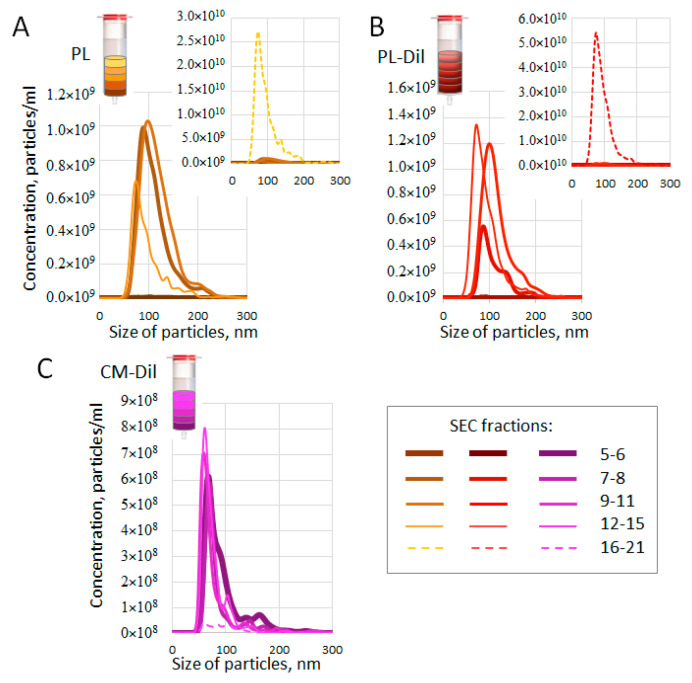
NTA analysis of SEC-fractionated samples. (**A**) Pellet pure plasma (2 mL); (**B**) pellet pure plasma (2 mL) stained with CM-Dil (2.5 fM); (**C**) CM-Dil suspension in PBS (2.5 fM). In all samples, SEC fractions were combined as shown in the legend and analyzed.

**Figure 5 membranes-11-00526-f005:**
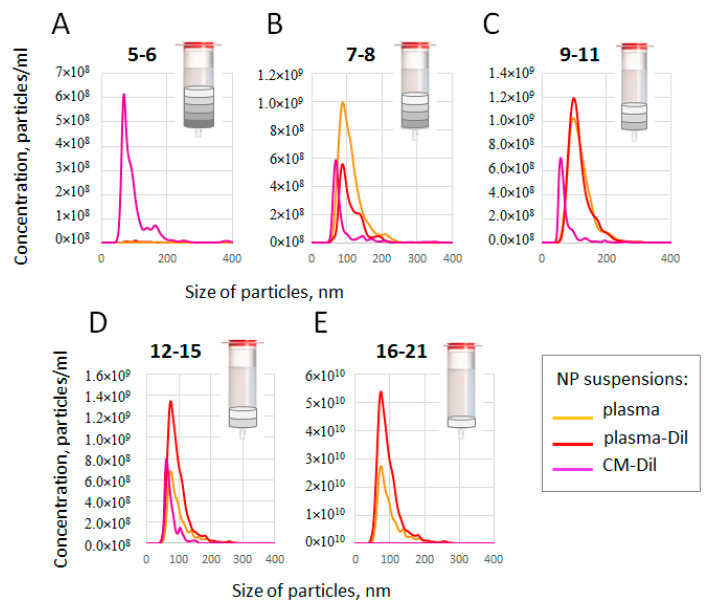
Comparative presentation of different SEC fractions measured via NTA. (**A**) SEC fractions 5 and 6; (**B**) SEC fractions 7 and 8); (**C**) SEC fractions 9–11; (**D**) SEC fractions 12–15; (**E**) SEC fractions 16–21. Line colors reflect nanoparticle suspension samples fractionated by means of SEC and analyzed via NTA.

**Figure 6 membranes-11-00526-f006:**
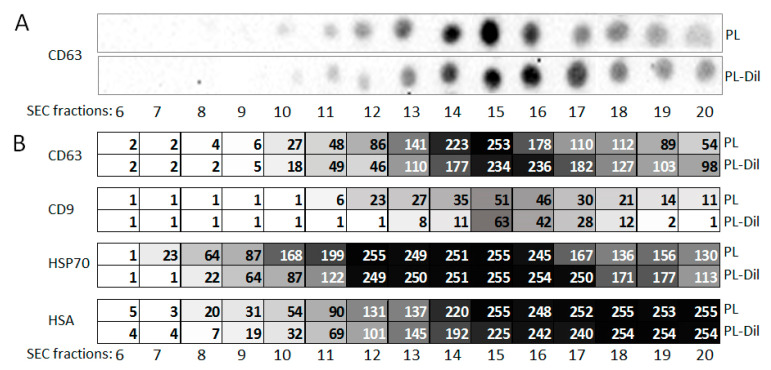
Dot-blot analysis of SEC fractions of plasma and CM-Dil-stained plasma. Particle suspensions obtained via fractionation were spotted, stained with antibodies, and their signal intensity was evaluated with HRP-labeled secondary antibodies and HRP substrate. (**A**) Dot-blot images after CD63 analysis; (**B**) results of all data quantifications with the Invitrogen iBright FL1500 Imaging System.

**Figure 7 membranes-11-00526-f007:**
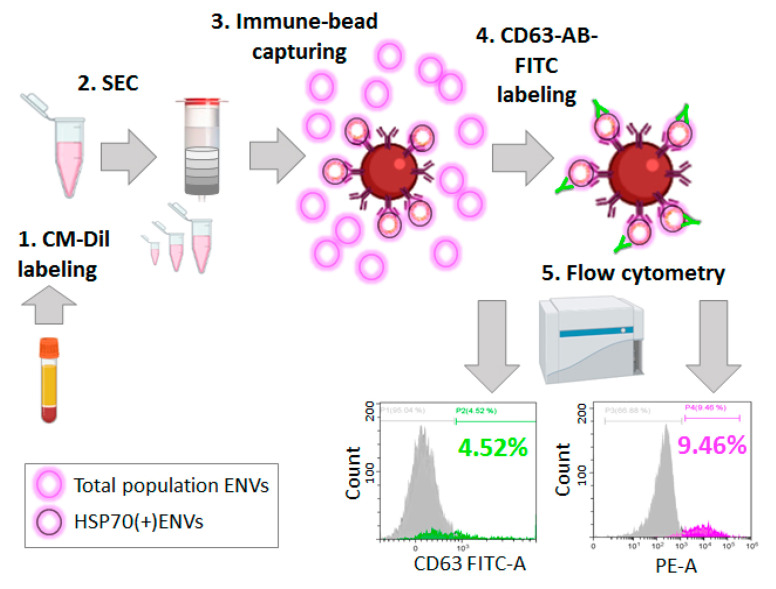
Schema of semi-quantitative analysis of HSP70(+) ENVs in plasma. Workflow includes (1) CM-Dil labeling of pellet pure plasma; (2) fractionation of CM-Dil-stained plasma by means of SEC and isolation of ENV-enriched fractions (fractions 9–11); (3) immune-affine capturing of HSP70(+) ENVs by immune-beads; (4) labeling captured HSP70(+) ENVs by antibodies against CD63 coupled with FITC; (5) flow cytometry in two channels to obtain a reading of the FITC signal (525 nm) and the CM-Dil signal (575 nm) in parallel.

**Figure 8 membranes-11-00526-f008:**
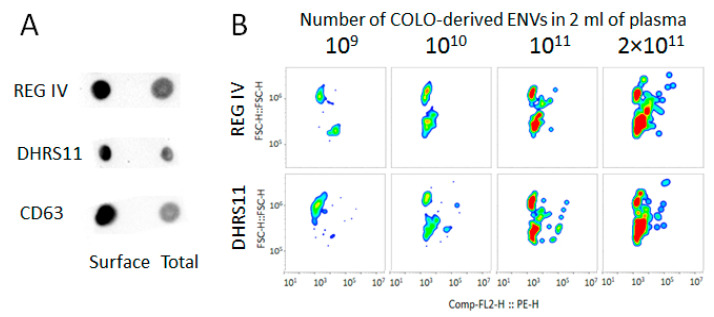
Semi-quantitative analysis of COLO320-derived ENVs in plasma. (**A**) Images of dot-blot analysis of intact vesicle suspension (Surface) and RIPA-lysed vesicles (Total); (**B**) results of bead-assisted FCM. The construction of pseudo-color/smooth plots (PE-H versus FSC-H) allowed us to detail the distribution of the fluorescence intensity among the positive events: SPMB with attached CM-Dil-labeled ENVs. The red-colored regions reflect the amount of highly fluorescent beads; green- and blue-colored regions reflect the distribution of beads with moderate and slight fluorescent signals, respectively.

**Table 1 membranes-11-00526-t001:** Content of the study.

Objectives	Methods/Readout	Analyzed Samples *			
		ENV/ CMdil	PPP/ CMdil	PPP -	- CMdil
1. Efficacy of labeling (SEC fractions 6–23)	1. Fluo-dot/fluorescent intensity assessment	+	+		
2. Nanoparticles’ characteristics (SEC fraction 5–21)	2. NTA/particle size and concentration.		+	+	+
	3. Dot blotting/specific protein content (CD9, CD63, HSP70, albumin)		+	+	
3. Evaluation of on-bead FCM sensitivity (SEC fractions 9–11)	4. On-bead FCM of samples labeled by two alternative approaches/fluorescent intensity assessment		+		
	5. On-bead FCM of samples containing different amount of ENVs of interest/fluorescent intensity assessment		+		

* ENVs/CM-Dil—plasma ENVs isolated by ultracentrifuge and stained with CM-Dil; PPP—pellet pure plasma; PPP/CM-Dil—pellet pure plasma stained with CM-Dil; CM-Dil—dye solution in PBS.

## Data Availability

Raw data on NTA are available in the Appendix A; other results are either presented in the article or can be provided upon request.

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
