# Peer review of "CM-Dil Staining and SEC of Plasma as an Approach to Increase Sensitivity of Extracellular Nanovesicles Quantification by Bead-Assisted Flow Cytometry"

_membranes, 2021, doi:10.3390/membranes11070526_

Round 1
Reviewer 1 Report
In the manuscript, CM-Dil staining and SEC of plasma аs an approach to increase sensitivity of extracellular nanovesicles quantification by bead-assisted flow cytometry, authors tried to address the limitations in quantification of EVs, as EV research is growing exponentially. However, i suggest few points to be considered before publication.
Materials and Methods:
2.4. CM-Dil plasma labeling and SEC purification, this should be present prior to NTA. As, sequence of events looks first labelling then purification and analysis.
Results:
-Authors did not perform TEM analysis of the isolated EVs, it is highly recommended to perform the TEM to show the isolated EVs were pure and not contaminated with other plasma components.
-As CM-Dil dye is lipophilic in nature, how did the author confirmed that CM-Dil dye did not label other plasma lipid component like lipoproteins? Please clarify.
-It is recommended to compare with protein binding dye as control like (Cy 3 NHS, Cy 5.5 NHS, FITC and so on) to verify the separated EVs reported to have EV proteins.
-Please elaborate the caption for Fig. 1, to make easy to the reader according to the sequence of events and same for Fig. 6.
Author Response
Dear Reviewer! We are very grateful for the careful analysis of our work and important comments made. We have made the required changes, and the specific questions are addressed below.
Materials and Methods:
2.4. CM-Dil plasma labeling and SEC purification, this should be present prior to NTA. As, sequence of events looks first labelling then purification and analysis.
Thank you for this comment. Done.
Results:
-Authors did not perform TEM analysis of the isolated EVs, it is highly recommended to perform the TEM to show the isolated EVs were pure and not contaminated with other plasma components.
In according to the recommendation, we performed cryo-TEM of ENVs isolated by UC. Since the concentration of ENVs isolated by SEC was about 3–4 orders of magnitude lower than UC-isolated ENVs, these samples were not suitable for Cryo-TEM analysis. Obtained images confirmed vesicular structure of UC-isolated particles. Data are included in revised manuscript.
-As CM-Dil dye is lipophilic in nature, how did the author confirmed that CM-Dil dye did not label other plasma lipid component like lipoproteins? Please clarify.
CM-Dil definitely labels ALL lipid-containing plasma components as it was estimated in the first experiment by fluo–dot. The main reason of size-exclusive chromatography is the isolation of ENV-enriched fractions from other plasma components. As it was shown in couple of reports, SEC allows to separate ENVs from the main classes of lipoproteins (Anita N. Böing et all., J Extracell Vesicles. 2014; 3: 10.3402/jev.v3.23430). If plasma was first stained with CM-Dil and then fractionated by SEC, it allowed us to isolate CM-Dil-stained ENVs with minimal contamination by CM-Dil-stained lipoproteins. This aspect is discussed in Discussion section.
-It is recommended to compare with protein binding dye as control like (Cy 3 NHS, Cy 5.5 NHS, FITC and so on) to verify the separated EVs reported to have EV proteins.
Thank you for this recommendation. We actually evaluated contain of CD9, CD63, HSP70 (EV markers) and albumin by dot-blotting (Figure 6). Theoretically, any method of non-specific labeling of plasma components followed by SEC-based isolation of ENV-enriched fraction can be established. However, taking in account known structure of ENVs (bi-lipid membrane vesicle), we suppose that non-specific labelling of lipids will be more efficient compare to labelling of proteins, nucleic acids or other components.
-Please elaborate the caption for Fig. 1, to make easy to the reader according to the sequence of events and same for Fig. 6.
Thank you for this comment. Done.
Reviewer 2 Report
It is an interesting work with important information concerning the extracellular nanovesicles quantification by bead-assisted flow cytometry, with possible applications in clinical practice.
A table - scheme describing the experimental sequence is needed to help the readers follow the experimental design and the results, due to the extensive methods and results description. Perhaps also a scheme with the final guidelines how to adopt the method would be really useful.
The last paragraph of the results concerning the application of the method on colon cancer should consist a separate chapter, as it is very interesting for clinical purposes.
The first paragraph of the discussion is a repetition of the results. It should be replaced by a more discussion-like content.
There are minor spelling mistakes.
Author Response
Dear Reviewer! We are very grateful for the careful analysis of our work and important comments made. We have made the required changes, and the specific questions are addressed below.
A table - scheme describing the experimental sequence is needed to help the readers follow the experimental design and the results, due to the extensive methods and results description. Perhaps also a scheme with the final guidelines how to adopt the method would be really useful.
We introduced additional paragraph in Results section describing design and content of study, this new paragraph was supplied by Table 1. The table summarizes objectives, methods and samples tested in the study. We also extended description to last scheme (Figure 8 now), that may now serve like guidelines how to use proposed approach.
The last paragraph of the results concerning the application of the method on colon cancer should consist a separate chapter, as it is very interesting for clinical purposes.
Thank you for this comment! It’s actually way we spent so many force and time to characterize proposed methods in detail. Currently, we expanded this approach and are analyzing big group of samples from patient with colon cancer using described here method of analysis. So, we decided to include this results as an illustration of usability of method and we would prefer to not expand this paragraph or present as separated chapter in this just methodological article.
The first paragraph of the discussion is a repetition of the results. It should be replaced by a more discussion-like content.
Thank you. We extended this section by discussing of possible applications of developed method.
There are minor spelling mistakes.
Thank you. The text was checked again and mistakes were corrected.